# Biological Rhythm and Chronotype: New Perspectives in Health

**DOI:** 10.3390/biom11040487

**Published:** 2021-03-24

**Authors:** Angela Montaruli, Lucia Castelli, Antonino Mulè, Raffaele Scurati, Fabio Esposito, Letizia Galasso, Eliana Roveda

**Affiliations:** 1Department of Biomedical Sciences for Health, University of Milan, Via G. Colombo 71, 20133 Milan, Italy; angela.montaruli@unimi.it (A.M.); lucia.castelli@unimi.it (L.C.); antonino.mule1@unimi.it (A.M.); raffaele.scurati@unimi.it (R.S.); fabio.esposito@unimi.it (F.E.); eliana.roveda@unimi.it (E.R.); 2IRCCS Istituto Ortopedico Galeazzi, Via R. Galeazzi 4, 20161 Milan, Italy

**Keywords:** chronotype, circadian typology, circadian rhythm, rest–activity, sleep, social jet lag, health, chronic diseases, melatonin

## Abstract

The circadian rhythm plays a fundamental role in regulating biological functions, including sleep–wake preference, body temperature, hormonal secretion, food intake, and cognitive and physical performance. Alterations in circadian rhythm can lead to chronic disease and impaired sleep. The circadian rhythmicity in human beings is represented by a complex phenotype. Indeed, over a 24-h period, a person’s preferred time to be more active or to sleep can be expressed in the concept of morningness–eveningness. Three chronotypes are distinguished: Morning, Neither, and Evening-types. Interindividual differences in chronotypes need to be considered to reduce the negative effects of circadian disruptions on health. In the present review, we examine the bi-directional influences of the rest–activity circadian rhythm and sleep–wake cycle in chronic pathologies and disorders. We analyze the concept and the main characteristics of the three chronotypes.

## 1. Introduction

Human nature has temporal components. Rhythm can be found at diverse organizational levels, from single cells to social behavior; indeed, nearly all physiological and psychological functions vary in periodicity. The most studied rhythms are the circadian rhythms, where circadian (from circa, about, and diem, day or 24 h) refers to functions whose cycle is around the 24 h. Circadian rhythms are generated by a core set of circadian clock genes that interact in a feedback loop and determine circadian periods and oscillations [1,2]. 

The central circadian clock that drives behavioral and humoral rhythms in humans is located in the suprachiasmatic nucleus (SCN) of the hypothalamus, which directly receives ambient light–dark stimuli from the retina and correlates them with sleep timing preferences. Circadian oscillators can be found in cells throughout the central nervous system and most other body cells. These peripheral clocks are synchronized by the central SCN clock and genetically equipped to generate circadian rhythms [3,4].

The association between rhythm and certain diseases has led to the notion that rhythmic bodily function is a characteristic of good health and that circadian disruption is harmful to health [5,6,7]. In fact, several studies showed that circadian disruption might lead to metabolic diseases, such as diabetes, obesity [8], cancer [9], neurodegenerative disease [10], and adverse cardiovascular consequences, such as an increased risk of cardiovascular disease-related mortality and stroke-related mortality [11,12].

In normal conditions, the endogenous rhythm of the sleep–wake cycle is synchronized with the alternation of the day–night cycle and other factors, such as the timing of meals and social routines (Figure 1) [13]. Such synchronization is essential to maintain healthy sleep–wake patterns as disruptions can lead to the emergence of different sleep problems [14,15].

The circadian rhythmicity in human beings is represented by a complex phenotype derived from multiple underlying genetic factors that define the chronotype. Regarding the intra-individual differences in the timing of the rhythm in bodily functions, inter-individual differences can be observed between persons who become active early in the day and those who do so later in the day. This trait, respectively termed *morningness* and *eveningness*, needs to be assessed not only as such but also in how it modifies the response to external schedules. Based on one’s intrinsic circadian rhythm, individuals differ in their preferred timing of sleep and activity; this may be expressed in the concept of chronotype [16].

Three different chronotypes are distinguished: Morning-types (M-types) and Evening-types (E-types), both of which are subdivided into extreme and moderate types, as well as Neither-types (N-types). A person’s chronotype lies on a continuum between morning and evening chronotype. Individuals with no pronounced circadian preference are categorized as N-types because they show intermediate characteristics. 

About 60% of the adult population is classified as N-types, and the remaining 40% in one of the other two [16,17]. E-types differ from M-types in their melatonin profile. Melatonin is a hormone produced by the pineal gland that influences behavior and physiology through its rhythmic synthesis. The pineal hormone is the best predictor of sleep onset [18]. It behaves like an endogenous rhythm synchronizer and promotes sleep through vasodilatory effects producing a fall in core temperature.

M-types and E-types differ in sleep–wake timing and mental–physical activation over a 24-h period. M-types go to bed and wake up early and achieve their peak mental and physical performance in the early part of the day [19], whereas E-types get up and retire later and reach their best performance during the second half of the day [16,17]. Chronotype can also influence attitudes, lifestyle, cognitive function, athletic performance, and personality traits [20,21,22,23,24,25]. M-types have been suggested to be more conscientious, agreeable, and achievement-oriented. In contrast, E-types have been indicated to be slightly more extroverted, exhibit neurotic traits, and are more disposed to mental or psychiatric, mood, personality disturbances, and eating disorders [26,27,28,29,30,31,32,33].

In the current review, we analyzed the concept and the main characteristics of the three chronotypes and examined the bi-directional influences of the rest–activity circadian rhythm and sleep–wake cycle in chronic pathologies and disorders.

## 2. Materials and Methods

### 2.1. Eligibility Criteria

An extensive literature search was performed using recognized life science and biomedical electronic databases and manually searching reference lists of those articles found, specifically investigating chronotype, circadian rhythm, rest–activity, sleep, chronic disease, and melatonin. No language, publication date, or publication status restrictions were imposed. Even though there were no language restrictions, relevant previous studies were found only in English. This search was applied to the following electronic databases: Medline (1966–Present), Google Scholar (all electronic resources to date), and PubMed (1996–Present). The last search was run on 15 January 2021.

### 2.2. Information Sources and Search Strategy

Two members of the project team developed the search strategy. The keywords circadian rhythm, circadian typology, chronotype, health, chronic diseases, sleep behavior, social jet lag, rest–activity, circadian rhythm, and melatonin were combined using #AND and #OR. Journal articles, conference proceedings, and clinical reports were reviewed for potentially eligible studies. The reference list of all articles was further searched for additional publications.

### 2.3. Study Records (Selection and Data Collection Process)

Subjects of any age, gender, and medical condition were included as well as articles involving animal studies. The article titles and abstracts were screened for relevance. The full text of all potentially relevant studies was reviewed to determine whether it met the inclusion criteria.

Articles describing changes in chronotype across the lifespan, effects of disrupted rest–activity circadian rhythm and sleep on health status, social jet lag, and interrelations between chronotype and melatonin were included. Studies without proper data presentation, with unclear or vague protocol description, and without an in-depth discussion were excluded from this review.

## 3. Results

### 3.1. Study Selection

This review retrieved 2103 articles (Figure 2). After the initial screening, 911 were duplicates and removed; 689 were excluded as a non-relevant topic; 503 full-text articles were assessed for eligibility; and 286 were imprecise in protocols, methods, or data presentations. The number of final studies included in the review is 217. All 217 studies included in the review contained original data and were published in English. Hence, we continue to describe these results qualitatively.

### 3.2. Circadian Rhythm and Sleep–Wake Alterations

#### 3.2.1. Circadian Rhythm Assessment

A rhythm is a regular periodic component in a time sequence, summarized through three parameters: acrophase, amplitude, and Midline Estimating Statistic of Rhythm (MESOR). The Cosinor method estimates the rhythm parameters, which describes the circadian rhythms with an algorithmic pattern or sinusoidal waveform [34].

Acrophase is a measure of timing, and it corresponds to the phase angle of the crest of a single best-fitting cosine [35,36]; amplitude is the measure of one half the extent of rhythmic change in a cycle estimated by the sinusoidal function used to approximate the rhythm [37,38]; and MESOR (Midline Estimating Statistic of Rhythm) is the value midway between the highest and lowest values of the function used to approximate a rhythm [39].

Circadian rhythms of activity levels can be assessed through an objective approach and method: actigraphy [40]. Actigraphy is based on the use of an actigraph, a device containing a piezoelectric triaxial accelerometer that is able to record movement during the day and sleep parameters during the night [17,41,42,43,44]. An actigraph is similar to a watch and worn on the wrist of the non-dominant hand for at least 7 continuing days. After the collection period, specific actigraph software analyzes the data and provides a detailed and objective portrait of an individual’s circadian rhythm [45,46].

In addition to the Cosinor, other methods have been developed over the years, such as the Hidden Markov Model [47], Dichotomy index [48], and the Square-Wave methods [49]; nonetheless, the majority of the studies examined in the present review utilized the Cosinor method.

#### 3.2.2. Circadian Rhythm and Sleep–Wake Alterations in Aging

Circadian rhythms are affected by age. Abnormalities of circadian rhythms in the elderly are hypothesized to reduce daytime function and increase sleep problems [50]. Specifically, diminished circadian amplitude and circadian acrophase lability tend to occur in the early stages of advancing age. Specifically, reduced circadian amplitude and circadian acrophase lability tend to occur in the early stages of advancing age. 

These alterations show inter-individual differences, and past disease history could worsen the aging-related circadian disruptions [51]. Similar findings were observed in the experimental conditions after bilateral lesioning of the SCN [52], suggesting the contribution of clock genes during the aging process [51]. Circadian disruption is detrimental to well-being. Consequently, metabolic disease, such as obesity and diabetes could be exacerbated by pre-existing circadian rhythm alterations [8].

Aging affects the sleep parameters, and in particular there is a decrease in slow-wave sleep, actual sleep time, and efficiency, as well as an increased number and duration of nighttime wakening, sleep latency, and fragmentation [50,53,54,55]. Aging is also characterized by decreased daytime activity levels and increased daytime naps [56,57,58].

#### 3.2.3. Circadian Rhythm and Sleep–Wake Alterations in Cancer

Rest–activity circadian rhythm abnormalities are observed in several chronic pathological conditions, such as cancer [9,59]. Quite a few clinical studies revealed the prognostic role of rest–activity and cortisol circadian rhythms in mortality rates among breast [60,61], colorectal [9,62], lung [63], and renal cancer patients [64].

Circadian disruption in cancer patients may have several etiologies. Both physiological effects and psychological distress are strongly related to cancer onset and abnormalities of circadian rhythm and sleep patterns [65,66]. Additionally, several studies assessed the mediating effects of endocrine and immune factors [67,68].

Both early and advanced cancer stages show marked circadian disruption affecting the rest–activity, metabolic, and immune cell rhythms [68]. Disrupted rest–activity rhythms co-occur with low physical function and low quality of life [69]. In such situations, central circadian disruption can enhance the cancer risk and hasten tumor growth and progression [68]. On the other side, medical treatments could negatively affect circadian rhythms. Sleep disorders, rhythm disruption, and fatigue, when already present before chemotherapy, are exacerbated by the treatments [70,71]. Specifically, the glucocorticoid rhythm represents the SCN’s key signal, and it controls the circadian clocks of peripheral tissues by regulating the proliferation of cells and the consequent cellular apoptosis, cell trafficking, and cytokine secretion [68,72].

Several studies show that sleep disorders are associated with an increased risk of cancer [73,74]. The associations between sleep duration (as a consequence of total or partial (2 h) sleep deprivation) and cancer are based on numerous mechanisms, including shifts in the aforementioned circadian system and in metabolism, immune, and endocrine function [75,76,77,78,79,80]. The mechanisms leading to the influence of sleep on cancer pathology final outcomes are not fully understood. Indeed, the diagnoses of cancer and/or cancer therapies are often linked to fatigue, depression, and pain that can intensify a pre-existing sleep disorder or participate in commencing a new one [81]. Chemotherapy or endocrine therapy are variables contributing to the high rates of sleep problems among cancer survivors, including the occurrence or exacerbation of menopausal symptoms (e.g., hot flashes) [82].

#### 3.2.4. Circadian Rhythm and Sleep–Wake Alterations in Neurodegenerative Disease

In neurodegenerative disease, rest–activity circadian rhythm abnormalities are observed in older adults with dementia, establishing an estimation of shorter survival (Gehrman et al., 2004) and increasing the mortality probability [10,11,12,83]. In addition to being recognized in dementia, low activity rhythms are particularly noticeable in Alzheimer’s disease. Lowered activity rhythms are described with reduced amplitude, phase delay of circadian variation in the core body temperature, and activity. These are all relevant factors for disturbances of the sleep–wake cycle [84,85,86,87].

Tranah and colleagues (2011) demonstrated that both late and advanced acrophase was combined with dementia or mild cognitive impairment in the elderly [10]. The mechanism behind this relationship is not clear; however, an altered circadian rhythm may influence several neurophysiological processes [10]. One possible explanation could include neuroanatomical and neurophysiological mechanisms under the control of the SCN pacemaker, and, through these two mechanisms, disorders in the circadian timing system affect memory, cognitive function, and behavior [88,89].

Adequate cognitive status depends on temporal alignment between sleep and clock-driven mechanisms. With aging, the local clock machinery’s compromise could reduce the temporal adaptation efficiency and impaired cognitive function. Indeed, multisynaptic pathways between SNC and the locus coeruleus’ noradrenergic neurons are neuroanatomical circuits mediating circadian regulation and influencing memory and cognition. More specifically, noradrenergic neurotransmission arising from the locus coeruleus can stimulate the frontal cortex and regulate arousal and weakness. Through this neuroanatomical circuit, for example, depression is associated with both decreased noradrenergic and serotonergic activity [90,91].

Sleep quality worsens with aging, both in terms of decreased duration and the inability to stay asleep at night [92,93]. Other complaints associated with neurodegenerative and cognitive diseases are insomnia, hypersomnia, parasomnia, nocturnal motor activity, and nocturnal dyspnea [94,95,96,97]. In this view, appropriate sleep quality and solutions to improve sleep quality are hypothesized to be useful in maintaining cognitive function in the elderly and reducing the most common cause of dementia, including Alzheimer’s disease. For example, sleep consolidation, reduced sleep debt, and improved sleep hygiene could attenuate the risk of neurofibrillary tangle density and improve the clearance of toxic accumulations of amyloid-beta [98,99].

#### 3.2.5. Circadian Rhythm and Sleep–Wake Alterations in Cardiovascular Disease

The elderly are usually more subject to adverse cardiovascular manifestations, such as the increased risk of cardiovascular disease (CVD) and stroke-related mortality. Causes for damaging cardiovascular disease could also be found in disruptions of the circadian activity rhythms [11,12]. More specifically, lower amplitude and higher minimum activity counts in older men were significantly and independently associated with a higher risk of CVD and coronary heart disease (CHD) events, as demonstrated by Paudel and colleagues (2011) [100]. 

On the other hand, a delay in the acrophase may increase the risk of peripheral vascular disease events. Concerning women, Tranah and colleagues (2010) reported that a reduced MESOR increased the risk of CHD-related mortality [11]. Acute CVD, such as the onset of myocardial infarction, transient myocardial ischemia, and atherosclerotic plaque rupture, were found to manifest a circadian component in the timing of occurrence. Muller and colleagues suggested a higher frequency of acute CVD during the early morning hours. These events could be related to the blood pressure’s acrophase, which occurs early in the morning [101].

In addition to other factors without any circadian influence (such as lifestyle or sedentary habits), some biological mechanisms lie behind the reciprocal influences between disrupted rest–activity circadian rhythms and increased CVD risk. The relationship could be defined in a bi-directional way, and circadian rhythm disruptions generally appeared to anticipate CVD-related events; on the other hand, it is conceivable that prevalent CVD disease worsens circadian rhythm disruptions by the debilitating impact on sleep–wake activity [100].

Interestingly, sleep duration was recently associated with vascular damage. In general, the recommended sleep duration is between 7 and 7.5/8 h, and its elongation or shortening could lead to a higher risk of mortality for specific pathologies. Indeed, sleepers who habitually have a long (>8–9 h) or short (<7 h, but more typically 5 h) duration of nocturnal sleep demonstrated increased risk to develop calcification of the coronary arteries [102], coronary heart disease [103,104], type 2 diabetes [105,106,107,108], stroke [106], atherosclerosis [109], and death [106,110,111]. Additionally, coronary heart diseases are more frequent in subjects when coupled with altered sleep duration and disturbed sleep [102].

The mechanisms that trigger the association between irregular sleep and cardiovascular events are not fully understood. A possible explanation linking sleep irregularities to CVD risk is that sleep deprivation generates a hormonal imbalance between the circulating leptin and ghrelin levels [112,113], which increase appetite, caloric intake, and reduce energy expenditure [114]. This situation leads to the condition of obesity and impaired glycemic control [115] with a resultant rise of cardiovascular risk. Short sleep activates low-grade inflammation with possible implications for cardiovascular disease and cancer diseases [116].

#### 3.2.6. Circadian Rhythm and Sleep–Wake Alterations in Eating and Metabolic Disorders

Eating disorders and obesity represent a source of great concern in modern society. Eating habits could have a relationship with circadian rhythm disorders. On this basis, Cornelissen and Otsuka (2017) investigated how meal frequency and timing could affect circadian rhythms and concluded that food restriction, such as a single daily meal, could amplify the circadian rhythm and increase the amplitude in mice [51].

However, the disruption of circadian rhythms is not the only factor linked to obesity and poor eating behavior. Evidence in many populations, including the elderly, demonstrated the link between short sleep duration and obesity [117,118]. Specifically, a study showed that irregularity in night-to-night sleep duration and a more considerable amount of time spent napping during the day were deeply related to obesity [119]. In this way, the relationship between body mass index (BMI) and sleep behavior is essential and bidirectional, and obesity could enhance the risk of sleep disorders. 

Likewise, restrictions in sleep duration can affect the body’s metabolic and nutritional balance [5]. Sleep–wake abnormalities may also lead to irregularity in eating behavior particularly in young adults. Baron’s study (2011) demonstrated that this population is characterized by dangerous behaviors, including later bedtimes, higher consumption of calories, and more frequent craving episodes [120]. Both snacking between meals and skipping breakfast predict increased body weight [121,122,123,124]. In addition to obesity, irregular eating behaviors are risk factors for metabolic syndrome [125].

Among the eating disorders that characterize today’s society, Binge Eating Disorder (BED), is a condition that could expose subjects to a higher risk of developing abnormalities of the rest–activity circadian rhythm. As demonstrated by Roveda and colleagues (2018), BED patients showed significantly lower actigraphic MESOR and amplitude values compared with the control group. A possible explanation may lie in the relationship between rest–activity circadian rhythms and abnormal eating patterns; thus, overeating and altering long-lived food behavior in BED patients may generate a circadian amplitude reduction [5].

The risk of eating disorders could also be related to low sleep quality, even though this relationship is not understood in depth. Several studies specifically examined the connection between sleep and binge eating disorders. Roveda and colleagues (2018) demonstrated that sleep debt in patients with BED was less dependent on eating disorders rather than obesity and lower physical activity levels [5]. Previously, Vardar and colleagues (2004) compared sleep in obese healthy subjects, obese patients without BED, and BED patients. Independent from the obesity status, the subjects without BED reported a higher subjective quality of sleep and lower sleep latency compared with BED subjects [126]. In agreement with these results, other studies reported similar findings [127,128,129].

Aspen and colleagues (2014) assessed interesting interactions between the appetite-regulating hormones ghrelin and leptin (which work in tandem to signal hunger and then promote satiety), disrupted sleep, and eating disorders [130]. More specifically, the sleep deprivation associated with decreased leptin and increased ghrelin levels [112,131,132,133] was found to generate a hormonal imbalance, and this condition may lead to weight gain due to increased appetite and reduced energy expenditure [134,135,136].

### 3.3. Chronotype, Sleep and Social Jet Lag Aspects

#### 3.3.1. Chronotype Assessment

Several self-assessment questionnaires are validated to assess chronotype. The most widely used and cited are: the Morningness–Eveningness Questionnaire (MEQ) [137], the Diurnal Type Scale (DST) [138], the Circadian Type Questionnaire (CTQ) [139], the Munich Chronotype Questionnaire (MCTQ) [1], and the most recent Morningness–Eveningness-Stability-Scale improved (MESSi) [140].

Actigraphy is an alternative and objective approach to assess chronotype [40,141]. Chronotype is directly correlated with actigraphic activity levels, based on differences in acrophase activity between chronotypes. While MESOR and amplitude showed no differences among chronotypes, the acrophase in E-types occurred nearly 2 h later than in M-types, which are more active in the early afternoon compared with E-types that peak in activity in the late afternoon [142]. 

Even though it is only about 2 h, the acrophase difference between the two chronotypes results are statistically significant. More specifically, the difference in the acrophase timing was about 2:20 h. Other authors, such as Lee and colleagues (2014), reported a similar rest–activity acrophase difference (approximately 2 h) [143]. The same different peak-times were found in other marker rhythms, such as temperature (2:10 h; 01:08 h) [144,145], cortisol (0:55 h), and melatonin (3:00 h) [18,144,145]

In conclusion, the activity acrophase derived from actigraphy data provides a new methodological approach to determining marker rhythm. Recent studies by Roveda and colleagues (2017) and Montaruli and colleagues (2017) investigated a simple linear model to predict the actigraphy-based acrophase from the scores of MEQ and the reduced Morningness Eveningness Questionnaire (rMEQ), which is an abbreviated questionnaire derived from the longer MEQ. 

The results support the assumption that the acrophase is linearly and inversely related to the MEQ and rMEQ score. Specifically, for each additional point in the MEQ score, the acrophase is about 5 min earlier, as well as for each additional point in the rMEQ score, the acrophase is about 16 min shorter. Simple, fast, and convenient, this method may prove useful when the actigraphy-based measurement of the acrophase is not applicable because it is too complicated, costly, or time-consuming [146,147].

#### 3.3.2. Chronotype: Influences of Gender and Age

Individual and environmental factors determine a person’s chronotype. The chronotype can also change with age and appears to be gender-related (Figure 1). In adults, eveningness predominates among males and morningness among females, although the difference disappears in menopause and during old age when early chronotypes are associated with early wake-up timing [148,149,150,151,152]. 

Not all studies found gender differences, and others reported evening characteristics as more prevalent in females. In adolescent samples, more E-types and later sleep timing were found for boys compared with girls [153,154]. Research also indicates possible gender-related differences in the reasons to get up, suggesting that grooming routines, household chores, or both could force girls to wake up earlier than boys on weekdays. Carskadon and colleagues (1993) find a correlation between later wake-up time on weekdays and eveningness among boys but not among girls [155].

The changes in the propensity toward preferring the morning or the evening that occur with advancing age are perhaps due to changes in gonadal hormone secretion [143]. Generally, children and the elderly are more morning-oriented, while younger adults are more evening-oriented [17,156]. Children are predisposed toward morningness until the start of adolescence, when preferences shift toward eveningness, whose peak is placed at the end of adolescence [155,156,157].

Numerous studies have assessed chronotype in schoolchildren, adolescents, and adults; however, research on Morningness–Eveningness in early childhood is scarce. Compared to preadolescents, adolescents tend to stay up progressively later, sleep later in the morning, and extend sleep on the weekends, as reported in an Italian study by Giannotti and colleagues [158]. Psychosocial changes between preadolescence and adolescence are thought to be the cause of the late sleep timing; these changes may include greater social opportunities, more academic responsibilities, and more possibilities for extracurricular activities and sports.

#### 3.3.3. Chronotype and Sleep

Chronotype is associated with differences in sleep–wake timing. Studies evaluating sleep quality in different chronotypes collected data via either self-assessment questionnaires or actigraphy [159]. Studies using self-assessment questionnaires showed that E-types were more prone to sleep complaints compared with M-types [33,160,161,162]. This condition was found to be already present at a young age (6–12 years) and during adolescence, with E-types showing difficulties in initiating and maintaining sleep [163,164]. E-types also reported suffering nightmares and insomnia symptoms more often than M-types [165].

Lehnkering and colleagues investigated the influence of chronotype on sleep behavior in young adults wearing wrist actigraphs and found a difference in the sleep efficiency between M-types and E-types [166]. Sleep efficiency is one of the sleep parameters evaluated by actigraphy, and this measures the percentage of time in bed actually spent sleeping. Both the use of self-assessment questionnaires and actigraphy to investigate differences in sleep parameters (e.g., sleep timing) between weekdays and weekends revealed that E-types go to bed and wake up much later than M-types on both work and non-workdays, and that eveningness is associated with later bedtimes and wake-up times and shorter times in bed during the week [158,167,168]. 

A study by Vitale and colleagues (2015) demonstrated that sleep quality and quantity were lower in E-types compared with in M-types on weekdays and that E-types reached the same levels as the other chronotypes on the weekend [142]. These data suggest that E-types accumulate a sleep debt during the week due to social commitments that force them to wake up earlier than their preferred times, and then they recover the debt over the weekend when they sleep better and longer.

Several recent studies regarding chronotype and sleep involve student populations, in particular university students. Silva and colleagues (2020) stated that E-type university students are typically characterized by poor sleep quality and higher anxiety traits compared to their M-types colleagues. Possible explanations could be academic demand in addition to the circadian desynchronization due to university schedules [169], and similar conclusions were previously reported in other studies [170,171]. In the same way, E-types workers forced to be active in early hours report poor sleep quality [172].

#### 3.3.4. Chronotype and Social Jet Lag

Jet lag generally refers to a misalignment between a person’s internal clock and external time cues after crossing multiple time zones. Inter-individual differences may differentially affect various circadian rhythms, including those associated with the adrenal cortex shift with the alternation of activity and rest [173], while those related to the pineal gland may be synchronized more directly with the alternation of light and darkness. The phase shift in endogenous melatonin secretion is correlated with sleep disturbances. Jet lag sufferers report impaired function or being out of synchrony due to the phase shift. Both extreme chronotypes suffer from jet lag although it can be said that M-types adjust their rhythm quicker compared with E-types [173].

To describe the difference in sleep timing between weekdays and rest days, Wittmann and colleagues (2008) introduced the concept of social jet lag, usually more frequently experienced by adolescents and E-types [174,175]. The sleep phase delay in adolescents on weekends is a well-known phenomenon; however, the problem appears to start already during the preadolescence phase [176]. In Italian elementary and junior high school children (age range 8 to 14 years) Russo and colleagues (2007) analyzed the relationship between sleep–wake habits, circadian preference, and self-reported sleep problems. 

They stated that the sleep–wake cycle delay starts during preadolescence, and bedtimes and rise times increase linearly with age on weekends, when parental control over bedtimes is less, and wake-up times are not dictated by school schedules [177]. These data, shared by findings for a transition to eveningness in American, Japanese, and Taiwanese children and adolescents, identified the shift toward eveningness at around age 13 years [178,179,180]. 

The differences between weekends and schooldays in bedtime, rise time, and total nocturnal sleep were more significant for E-types than for M-types. To recover their sleep debt accumulated during the week, E-types sleep more on weekends than on school days. The higher rates of sleep-related problems in E-types compared to M-types can be interpreted as a more pronounced misalignment between their biological rhythm and the social rhythm imposed by school schedules, and therefore E-types more frequently complain of daytime sleepiness [181].

Based on these findings, data on adolescent populations agree in indicating that adolescence is a stage in which the sleep–wake cycle tends to become delayed compared to the circadian phase position of the sleep–wake cycle in children. The differences in sleep habits between M- and E-types during adolescence may be influenced by developmental endocrine factors [182,183]. This could also be related to heavier academic and social demands, relaxed parental restrictions, and increased independence, as well as greater involvement in late-night activities. These findings suggest that the onset of adolescence is a time of physiological and social changes that affect sleep, decreasing sleep duration [184], and increasing sleep irregularity [158,177].

The evidence that many adolescents appear to obtain insufficient sleep is of particular interest from a health perspective. The growing recognition of the importance of sleep refers to physical health and cognitive and affective function [184]. The mismatch between internal timing and social schedules results in desynchronization of an individual’s habits: adolescents and young adults need to fit their chronotype around work/school schedules that impose wake-up times and activation timing contrary to their intrinsic circadian preference. 

A lack of synchronization with cultural standards related to activity timing can lead to worse academic achievement, higher anxiety, cardiometabolic risk, and complicated relationships with family members and school friends [26,169,179,185,186,187,188]. In this regard, the misalignment associated with eveningness can be partly considered a biological vulnerability factor concerning physical and cognitive performance as well as the development of psychological problems [20]. Conversely, morningness could be seen as a protective factor against adolescent psychopathology because it is associated with a lower risk for aggressive behavior, attention problems, and delinquent behavior [189,190].

However, social jet lag does not exclusively affect adolescents, as it has also been reported in shift workers, whose work during the night leads to a desynchronization of the circadian rhythm [191], and in young adults [192]. In contrast, not all populations showed a great social jet lag manifestation; for example, the Chinese population reported the social jet lag less frequently than the European population, and it was not correlated with higher BMI as it typically is in western societies [193]. Social jet lag represents a possible risk factor or cause for several pathological conditions, such as tumors, type 2 diabetes, and depression [194,195,196].

#### 3.3.5. Relationship between Chronotype and Melatonin

Melatonin, a hormone produced by the pineal gland in a circadian manner, has widespread effects on the body and acts as a neuroendocrine transducer and circadian signal [197]. Melatonin synthesis is under the direct control of the SCN. The synthesis lies through a sympathetic pathway that releases noradrenaline on the pinealocytes during the night, thereby, allowing the entrainment of the circadian rhythms of several biological functions [198]. 

Photic information from the retina is transmitted to the pineal gland by the SCN: the neural input to the gland is norepinephrine, and the output is melatonin. The autonomic nervous system, core body temperature, and melatonin rhythms are well-known output signals from the hypothalamus’ SCN to generate circadian rhythms in most physiological and behavioral variables [199]. However, because humans can voluntarily modify some of these output rhythms (e.g., meal-times, rest–activity times, or physical exercise), they can also act as input signals, which, in turn, modify the central and peripheral activity clocks. In other words, the innate and internal circadian system is influenced by external and variable factors.

In humans, melatonin secretion increases soon after the onset of darkness, with plasma concentrations reaching a night peak, which can be 30 times higher than daytime values, and falls during the second half of the night [17]. Endogenous melatonin release begins around 2 h before bedtime, toward dusk, and its production is inhibited by light [200]. The onset of melatonin secretion is recognized as the best predictor for sleep onset since it promotes sleep through vasodilatory effects that lower the core temperature [201]. 

The circadian rhythm for melatonin in humans is closely synchronized with regular sleep hours [202]. Desynchrony between the internal circadian clock and environmental light–dark conditions leads to mistimed melatonin release and sleep disruption [203]. A typical phase shift in the rhythm of endogenous melatonin secretion occurs in travelers after a flight across multiple time zones, as well as in shift workers [202,204].

Melatonin is used as a biological marker of neuroticism, introversion–extroversion, and Morningness–Eveningness [16,205]. M-types have an advance in the melatonin secretion rhythm phase compared to E-types: the onset, acrophase, and offset of the melatonin profiles occurs approximately 3 h earlier in M-types compared with in E-types, with no differences in amplitude [16]. Serum melatonin levels measured at 9:00 may be useful to differentiate M-types from E-types since E-types have a higher melatonin level at 9:00 than do M-types. 

However, the higher serum melatonin level recorded in E-types at 9:00 is not sufficient to induce a hypnotic effect on these subjects. The registered serum melatonin level is lower compared to the concentration required to induce a hypnotic effect. This difference may help to understand why E-types are sleepier and more awkward in the morning. At noon, the serum melatonin level typically does not show any differences between M- and E-types anymore [205].

The discrepancy between the timing in Dim Light Melatonin Onset (DLMO) and activity acrophases could be due to different period monitoring. Vitale and colleagues (2015) studied the period for the rest–activity circadian rhythm including both weekdays and the weekend days [142]. The imposition to follow social routine could have smoothed the difference between the two extreme chronotypes making the differences in rest–activity peaks less accentuated.

Maintenance of the circadian melatonin rhythm is considered a biological marker associated with successful aging, whereas flattening of this rhythm is associated with the appearance of neurodegenerative disease. The decrease in serum melatonin’s nighttime peak with advancing age indicates that melatonin plays a role in aging and age-related disease [202]. 

However, during aging, a decline in the amplitude of the melatonin curve is a characteristic of aging per se. Some conclusions of previous researches could explain the physiologic decline in melatonin’s amplitude. Indeed, the pineal gland responsible for melatonin secretion undergoes a calcification process with advancing age. The pineal gland’s calcification is associated with numerous neuronal diseases and pathological illnesses that are typical of the aging process [206].

Melatonin secretion may play various essential roles in human physiology and pathophysiology [207,208]. In addition to its hypnotic effects, melatonin is implicated in affective disorders, such as depression and seasonal affective disorders (SAD). SAD and mood disorders could be generated by circadian rhythm disfunctions and treated with chronobiotic drugs, chronotherapy, and bright light therapy. Melatonin has chronobiotic properties; however, there are different points of view regarding its role in the treatment of SAD and mood disorders. 

Even though melatonin is an effective treatment for sleep and circadian rhythm disorders, there no univocal results regarding its efficacy on SAD and mood disorders [202]. Serum melatonin assessment is gaining attention in pathological conditions, such as breast cancer, to evaluate sleep complaints and depression in patients [209].

#### 3.3.6. Chronotype and Chemotherapy Effects

Eveningness represents a risk factor for breast cancer, and this could be enhanced in combination with shift working [210]. Cancer survival and the related quality of life during therapy are gaining increasing attention in recent years. Studies that include chronotype assessment in cancer patients are still few and are mainly focused on women and breast cancer patients. A recent study indicated morning preferences as a protective factor against hot flashes due to menopause induced by chemotherapy treatments in breast cancer patients. The possible cause could be found in the body temperature’s circadian rhythm: E-types usually report a dysregulation in the hormones regulating reproductive functions and body temperature. This dysregulation could up-regulate the body temperature threshold, promoting more frequent and more invasive hot flashes [211].

Previously, Lee and colleagues (2017) investigated chronotype differences in nausea and vomiting manifestation in women experiencing the first cycle of the chemotherapy cycle. E-types report a higher rate of nausea and vomiting episodes compared to M-types. The authors’ explanation concerns the fact that certain neurotransmitters (i.e., monoaminergic neurotransmitters, such as hypocretin, gamma-aminobutyric acid, histamine, norepinephrine and 5-HT, substance P, and cholecystokinin) are involved at the same time with the circadian rhythm and with nausea and vomiting episodes; on the other hand, the timing of chemotherapy administration does not correspond to internal circadian timing [212].

In this view, proposals for circadian chemotherapy schedules have been advanced [213,214,215,216,217]. They are based on the evidence that the administered drug’s outcomes may change depending on the drug timing administration. Tissue susceptibility and reactions to toxicity (a consequence of anti-cancer drugs) also change during the circadian cycle. In this view, chemotherapy’s timing may lead to greater therapeutic specificity and lowered complications and toxicity [213]. For example, Levi and colleagues (1990) stated that two different chemotherapy drugs, 4’-O-tetrahydropyranyl doxorubicin and cisplatin, first administered in the morning and second, in the late evening, reduced the severity of the toxicity by two or three times [214].

## 4. Discussion and Conclusion

The data reported in this review highlight how alterations in circadian rhythms and the sleep–wake cycle are related to physiological processes, such as aging, and to pathological processes. The relationship between circadian rhythms and diseases could be described as bidirectional. In this view, circadian rhythm alterations could create predispositions to the development of several diseases (such as cancer and neurodegenerative, cardiovascular, and metabolic diseases) and to physiological aging; on the other hand, the presence of pathology is able to exacerbate circadian rhythm abnormalities. The most significant circadian rhythm alterations refer to the acrophase and amplitude of the biological variables and rest–activity circadian rhythms.

Sleep also acts in a bidirectional way in the development of chronic disease. Sleep disorders have detrimental effects on the sleep–wake cycle, which, in turn, creates predisposition to higher illness risks. However, the presence of disease could affect the sleep–wake cycle, worsening sleep quality. In this view, eveningness should be considered a risk factor that could compromise sleep.

Indeed, it has also been shown that circadian rhythmic expression can differ from person to person, and these individual differences in the chronotypes are able to influence both physiological and psychological functioning. Consequently, it is necessary to take into account that regular habits and social commitments tailored to the chronotype should be considered important factors that should not be underestimated in order to avoid desynchronization in the circadian rest–activity rhythm and in the sleep–wake cycle.

## Figures and Tables

**Figure 1 biomolecules-11-00487-f001:**
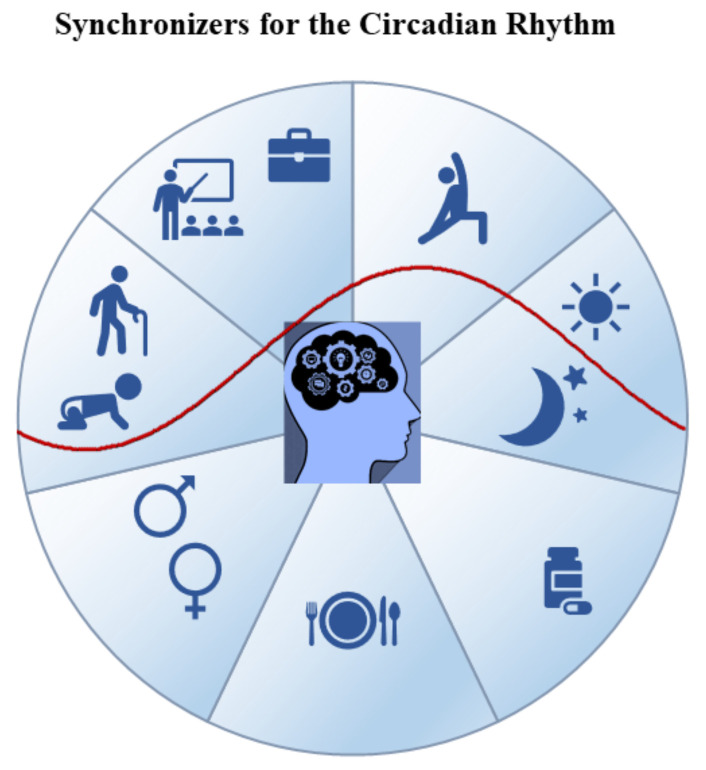
Factors influencing the rest–activity circadian rhythm and the sleep–wake cycle.

**Figure 2 biomolecules-11-00487-f002:**
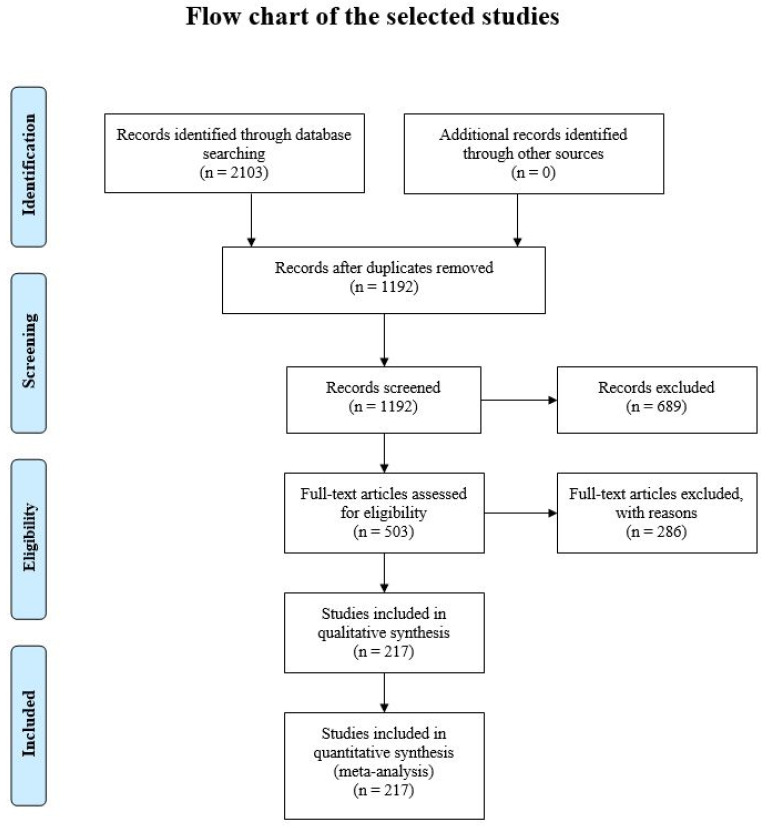
Flow chart of the selection of studies for inclusion in this review.

## Data Availability

Not applicable.

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
