# Peer review of "Biological Rhythm and Chronotype: New Perspectives in Health"

_biomolecules, 2021, doi:10.3390/biom11040487_

Round 1

Reviewer 1 Report

This paper is a review of the relationship between circadian rhythms, chronotype and aspects of human health. It lacks a critical perspective in terms of the studies reviewed and is too willing to accept at face value the conclusions of the authors of those papers. I wonder if the review is too broad in its focus and might have benefited from a more in-depth critic of the studies in a single area e.g., such as Cancer, circadian rhythms and chronotype. One of the difficulties in the paper is the conflation of chronotype, circadian rhythm and melatonin secretion. While they are all clearly related and it is difficult to discuss one without the other two, it may have been well to discuss this inter-relationship as an introductory section. [Just a suggestion not something necessarily to be adhered to in any revision submitted].  

I have some specific comments on the manuscript as follows:

I am not sure that figure 1 adds anything in the way of explanation to the article and could be omitted.

It is stated that "No language, publication date, or publication status restrictions were imposed" yet the papers retrieved appear to be all in English (line 112) and with a limit from 1966?

Section 3.2.1: is the detail here necessary at all since it is mostly standard textbook material. Perhaps simply state what the parameters are e.g., amplitude mesor etc and refer readers to an appropriate reference. Alternatively use a Text Box to define the various terms but do not include in the main article.

Line 139: Figure 1 does not give any explanation as to the mechanism of how age affects circadian rhythm it is simply a diagrammatic representation. 

Line 140: 'supposed' or 'have been demonstrated' or 'hypothesised'???

Line 141: no apostrophe in 'acrophase'

Line 142: rather vague. Do you mean from 65years onwards or is it more likely that this lability will exhibit high inter-individual variability as well as be dependent on the general state of health of an individual? Perhaps issues related to general physical health (diabetes cardiovascular disease etc) as well as past history of mental health issues (previous depressive/anxiety episodes, substance use disorder) may play a role here. Further physical anatomical factors may also be important e.g., state of pineal calcification? 

Line 164: This does not seem to follow from the previous sentence to me. Clearly there are two potentially opposing factors at play here: 1 the chemotherapy/radiation therapy which is designed to halt tumor progression and 2 the circadian disruption which may (or may not) hasten progression. Clearly the balance between the two is at issue or indeed may not be an issue at all that is if the treatment is not disruptive of the circadian cycle.  

Line 172:  This seems contradictory to the previous statement in line 164 that circadian disruption can hasten tumor progression. 

Line 180: Not clear here what is meant. How does the association of dementia and sleep disturbance "prove" anything?

Line 182 Too many ideas in the sentence which obfuscates the meaning.

Line 189: what neuroanataomical mechanism is leading (sic) under the control of the circadian system? This does not seem feasible to me. The statement is vague and needs to be more specific: name the potential systems. 
What seems more likely to me is an alteration of the cortisol rhythm. There is epidemiological evidence of a relationship between depression and dementia and between depression and hyper-cotisolaemia. Perhaps that is one hypothetical mechanism leading to the development of both dementia and circadian instability. However the two processes could be independent or at least inter-dependent.

Line 199: it would seem speculative at this time that improving sleep quality would reduce dementia. Findings that amyloid-beta 42 protein is cleared from the brain during sleep certainly add some credence to this idea but further research would seem to be necessary before such blanket statements could be made. 

Line 210: again this statement seems somewhat vague more a 'maybe' situation. If this is a speculation then "might" is appropriate or if it has  been demonstrated then 'does' is appropriate.

Line 211-212: CVD is a state either you have it or you don't. It does not fluctuate on a daily basis. Yes it may improve or deteriorate over time but not across a day. In fact I would suggest you would be hard pressed indeed to show daily variations in the conditions of the coronary arteries even in an individual with severe disease within a single day. What do you mean here BP, HR,Mean arterial pressure some other parameter? Both HR and BP are well known to exhibit a circadian rhythm. In CVD do you mean the BP rhythm is de-synchronised from other rhythms or is different in CVD patients compared to controls. Do you mean the incidence of AMI (acute coronary syndrome) which are known to occur more frequenTly in the AM?

Line 215: While the relationship between circadian disturbance and CVD related events (stroke, infarct) may exhibit an association there are multiple factors which are related to the occurrence of such events, many of which are life-style factors without any circadian influence. 

Line 220-223: This seems incredible as the window is very tight (just 60minutes). The statement also seems to imply that those who sleep the requisite 8 hours do not die! I think you mean lower risk of mortality not actual mortality. 

Line 224 repetitive of line 213.

Line 225: These seem more plausible links to CVD risk factors than stated in line 213 through 218.

Line 291: rMEQ is not defined

Line 302: "and during the elderly" meaning???

Line 369 missing words in sentence

Line 416: melatonin is not a circadian clock it is a circadian signal, The clock is the SCN

Lines 424-427: In other words the circadian system is influenced by zeitgebers.

Line 438: The paper quoted here does not address the relationship between personality characteristics and melatonin secretion other than an un-referenced mention in the abstract. Instead it examines the relationship between melatonin and chronotype in a small cohort. It seems unlikely that melatonin is a a marker of traits such as neuroticism etc, at least not from this study.

Line 444: this would assume that melatonin in the concentrations present at 0900h is in fact hypnotic. Use of melatonin as a hypnotic in the clinic involves doses of 2-3mg and plasma concentrations which exceed physiological levels by more than 100-fold

line 448 A decline in amplitude of the melatonin curve is a characteristic of aging per se. What it may indicate is clacification of the pineal gland with age.

Line 452: More recent publications would suggest that any declines in melatonin secretion in depression are probably medication effects not an a priori relationship to causation of depression. In SAD there may be a role of melatonin as a treatment or for the use of bright light. 

Line 468 tautological

line 469 which neurotransmitters? dopamine?

Reviewer 2 Report

This review by Montaruli A. and Colleagues critically summarises the literature on circadian phenotype, circadian rhythms and various diseases.

The structure is altogether appropriate and novel, and the literature pertinent, and its focus rather novel in the context of several reviews on circadian rhythms and disease.

I have a few remarks, though, all rather minor:

Page 1, introduction: for clarity, the Authors ought to introduce the definition of circadian for non-expert readers;

Page 1, line 43: Reference #9 might not be correct for cardiovascular outcomes;

Lines 118-125 and 126-132 are a repetition and this whole paragraph could be simplified;

Line 133: the Authors ought to describe what Actigraphy is for non-expert readers;

Line 137: the Authors ought to discuss non-parametric tests for analysing acrigraphy time-series as cosinor method is far from being the most widely-used one;

Line 154: the Authors ought to specify which rhythms were assessed;

Line 158: reference missing after "patterns";

Line 160: instead of first, to the Authors mean early?

Line 162: reference missing after "life"; what do the Author mean by physical QoL? This is not too widely used;

Line 170: sleep duration is too vague: the Authors ought to specify whether too short, too long, or both?

Line 172-173: this is not correct: there are at least two papers, one using actigraphy and one using QoL questionnaire, linking sleep troubles with prognosis in cancer patients;

Line 181: reference number missing (only author, year provided);

Line 236: the Authors ought to disclose the outcome of the cited study, not just the setting of the investigation;

Line 254: do the Authors refer to actigraphy MESOR, or of another rhythm?

Line 285: the Authors ought to comment on the relatively small difference in acrophase (~2 hours) between E and M types, as other studies have found a markedly wider difference in, for instance, skin temperature acrophases between subjects;

Line 291: the Authors ought to define what rMEQ is for non-experienced readers;

Line 294: what does shorter mean for the acrophase? Earlier?

Line 357: reference missing after "rest";

Line 393: reference missing after "function";

Line 424: reference missing after "variables";

Line 432: reference missing after "temperature";

Line 441: the Authors ought to comment here, with regards to line 285, too, the discrepancy between a ~3-hour difference in DLMO times versus ~2-h in activity acrophase;

Line 449: the decrease in melatonin could either be a cause of age-related disease, or a consequence of it; the sentence is incorrect unless the Authors provide causative links, as association does not imply causality;

Line 473: the Authors ought to expand on circadian chemotherapy schedules: what are they? Are they different from chronomodulated chemotherapy? What is chronomodulated chemotherapy? The readers here are left with unexplained and undefined wordings, altogether unclear;

Lines 475-481 and 482-490 are basically a repetition, and the whole conclusion could be shortened and simplified.

Reviewer 3 Report

The manuscript entitle Biological rhythm and circadian phenotype: nw perspectives in ealth a narrative review, is confusing starting from the title, that no focused the aim of the review. What is a narrative review? In all the text there are some non correct affirmations.

Keywords are not appropriate.

Figure 1 and 2 are not clear and should be deleted.

Materials and method section is not appropriate for a review

Result section is not appropriate for a review 

It is not clear if the author want to review the literature about the role of circadian rhythm disruption on diseases, or if they want review the literature about the different phenotype in humans. In the first case the paragraph 3.3 should be deleted, in the second case the paragraph 3.2 should be deleted.

Discussion and conclusion should be focused on the chosen topic

Reviewer 4 Report

Title: Biological rhythm and circadian phenotype: new perspectives in health – a narrative review

Author: Montaruli et al.

This review article summarizes recent advances in circadian rhythm and propensity of health impairment. The topic with which this review article is dealing with is interesting and important. The method is sound and the result of literature search is properly displayed.

However, there is one thing which is not clear enough to get the concept of “circadian phenotype”. This reviewer recognizes the expression is used in a line of scientific literature which deals with genetically engineered mice as well as human alcoholics. So far no literature has discussed in detail the concept, circadian phenotype. This review should open such discussion; is this a matter of genetic background, hormonal changes, social pressure etc. In a similar line of points, if three chronotypes are innate, there would be nothing we can do to change E-type behaviors or tendency to show some health disorders. In any event, the word “innate” is ambiguous and misleading, which may mean genetic, apparent on birth, or something else. Therefore, the word should be replaced with more clearly defined one.

Round 2

Reviewer 3 Report

I am sorry, but the authors did not follow my suggestion. The manuscript in this version is not suitable for publication